# A minimal tensor network beyond free fermions

**Carolin Wille**[1★]**, Maksimilian Usoltcev**[2†]**, Jens Eisert**[3,4‡] **and Alexander Altland**[2∘]

**1** Department of Applied Mathematics and Theoretical Physics, University of Cambridge, Cambridge CB3 0WA, United Kingdom
**2** Institut für Theoretische Physik, 50937 Cologne, Germany
**3** Dahlem Center for Complex Quantum Systems, Freie Universität Berlin, 14195 Berlin, Germany
**4** Helmholtz-Zentrum Berlin für Materialien und Energie, 14109 Berlin, Germany

★ carolin.wille@fu-berlin.de
† usoltcev@thp.uni-koeln.de
‡ jense@zedat.fu-berlin.de
∘ alexal@thp.uni-koeln.de

## Abstract

This work proposes a minimal model extending the duality between classical statistical spin systems and fermionic systems beyond the case of free fermions. A Jordan-Wigner transformation applied to a two-dimensional tensor network maps the partition sum of a classical statistical mechanics model to a Grassmann variable integral, structurally similar to the path integral for interacting fermions in two dimensions. The resulting model is simple, featuring only two parameters: one governing spin-spin interaction (dual to effective hopping strength in the fermionic picture), the other measuring the deviation from the free fermion limit. Nevertheless, it exhibits a rich phase diagram, partially stabilized by elements of topology, and featuring three phases meeting at a tricritical point. Besides the interpretation as a spin and fermionic system, the model is closely related to loop gas and vertex models and can be interpreted as a parity-preserving (non-unitary) circuit. Its minimal construction makes it an ideal reference system for studying nonlinearities in tensor networks and deriving results by means of duality.

# 1  Introduction

The duality between the classical two-dimensional Ising system and free fermionic systems and its numerous generalizations have been a subject of intense research over several decades [1–15]. In previous work [16], we have expanded upon those dualities: concretely, the duality between the ground state of the toric code [17], that of a class D topological superconductor [18], and the partition sum of the classical two-dimensional Ising model, with a two-dimensional *tensor network* (TN) [19–22] playing the role of an efficient and intuitive translating element. For this to work, the tensor network has been chosen from the matchgate category, which offers a distinct advantage: a matchgate tensor network stands in a one-to-one correspondence with free fermionic systems [23–25]. Much like in the study of those, where rich physics is uncovered by adding electron-electron correlations, a natural extension is to consider a tensor network that feature non-linearities.

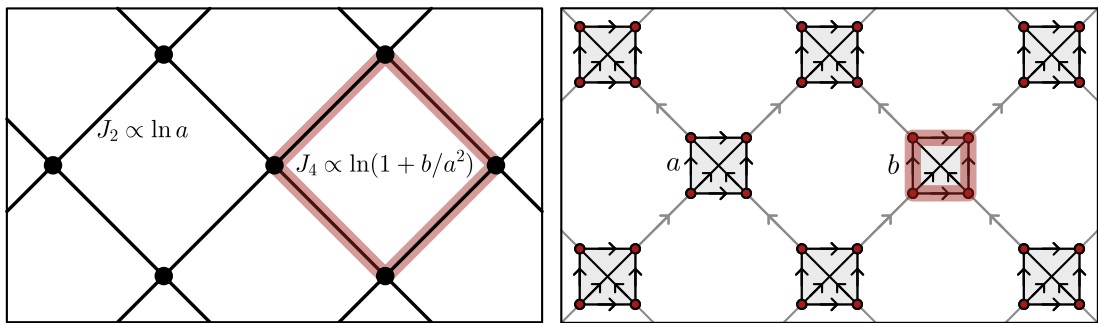

Figure 1: The model in its spin (left) and fermionic (right) version. Left: Classical spins are located on the lattice dual to the one on the left. They interact with nearest neighbour Ising interactions and a 4-spin plaquette term (red shading). Right: Fermionic modes are marked by red dots. Black (grey) arrows denote the 2-fermion intra (inter) unit cell couplings with weight $a$ (unity). The four fermion ring exchange term is indicated with red shading.

In this work, we set up a minimal model and study the effect of adding non-linearities in the fermion picture (i.e., going beyond matchgates) on the other side of the dualities. The model is defined in terms of two parameters, $a$ and $b$, interpreted differently depending on the perspective. In the context of a statistical mechanics model, $a$ describes classical nearest-neighbor spin interactions, while in the fermionic picture, it corresponds to an intra-unit-cell hopping strength. The parameter $b$ describes the strength of non-linearity. It is represented as a four-spin interaction in the statistical mechanics model or, in the fermionic model, as a four-fermion term, akin to a Majorana ring-exchange term. Concretely, the statistical mechanics model is characterized by a spin Hamiltonian

$$H_{\text{Ising+}} = -J_2 \sum_{\{i,j\} \in e} s_i s_j - J_4 \sum_{\{i,j,k,l\} \in p} P_4(s_i, s_j, s_k, s_l) \,, \tag{1}$$

where $e$ and $p$ denote the edges and plaquettes of a square lattice, respectively, $J_2 = -\frac{1}{2} \ln a$ and $J_4 = -\ln(1 + b/a^2)$ are the coupling strengths (cf. Fig. 1 left), and $P_4$ is a projector onto 4-

spin plaquette configurations in a checkerboard pattern (cf. last entry of Tab. 1). Meanwhile, the fermionic model is described by the (pseudo-) Hamiltonian

$$H_{\text{fermion}} = i\left(\frac{1}{2}\theta^{\mathsf{T}}(\otimes_x A + C)\theta + b\sum_x \theta_1^{(x)}\theta_2^{(x)}\theta_3^{(x)}\theta_4^{(x)}\right), \tag{2}$$

where $A_{i,j} = a\operatorname{sgn}(j-i)$ denotes the hopping strength within a unit cell, $x$ sums over all unit cells and $C$ represents the unit-strength hopping between different unit cells (cf. Fig. 1 right). For suitable boundary conditions explained below, the partition functions

$$Z_{\text{spin}} = \sum e^{-\beta H_{\text{Ising+}}}, \quad Z_{\text{fermion}} = \int d\theta\, e^{-iH_{\text{fermion}}} \tag{3}$$

coincide for all system sizes.

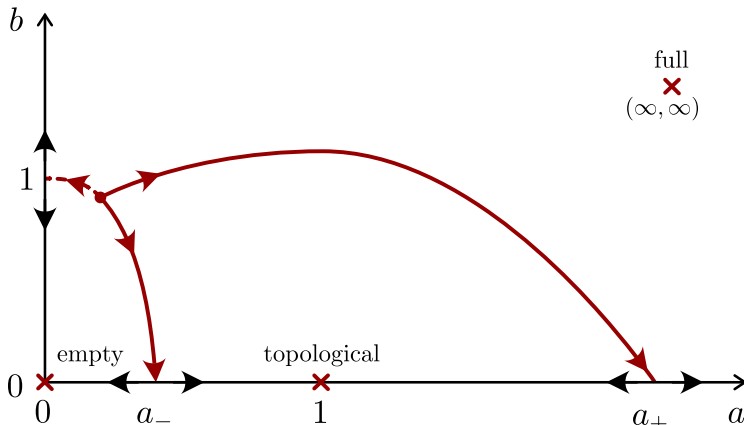

Figure 2: Phase diagram (qualitative). Solid (dashed) lines mark second (first) order phase transitions. Crosses mark the stable fixed points.

In the case without non-linearities, $b = 0$, the two-parameter model in the spin interpretation reduces to a classical Ising model with $a \propto \ln\beta J$ ($J$ is the coupling strength and $\beta$ inverse temperature). In the fermionic picture, $b = 0$ leads to a class D topological superconductor. These models, and, in particular, their phase diagrams are well-understood. The Ising model features a ferromagnetic, a paramagnetic and an anti-ferromagnetic phase, separated by transitions at $a_- = \sqrt{2}-1$ and $a_+ = \sqrt{2}+1$. By Kramers-Wannier duality, we can interpret the domain walls between 'up' and 'down' spins as strings and thus obtain a dual statistical mechanics model, a loop gas, where the partition sum runs over all closed loop configurations. It is in this picture that the non-linearity is most natural and most easily understood in qualitative terms — it amounts to favoring or disfavoring domain-wall crossings.

In this loop gas language, the three phases are referred to as 'empty', 'topological' and 'full', respectively. The name 'topological' in this context is justified by the fact that the dual of the paramagnetic Ising phase can be thought of as the classical analogue of an equal weight superposition of closed loops featured in the toric code ground state. It is notable that this 'topological' phase coincides with the topological single-particle phase in the fermionic version of the system, which manifests in the emergence of edge modes. In contrast, both analogues of the 'empty' (ferromagnetic) and 'full' (anti-ferromagnetic) phases exhibit trivial band topology as indicated by their zero Chern numbers.

In light of the topological features, it is perhaps unsurprising that the phases remains robust against the presence of weak non-linearities. However, for stronger non-linearities, the picture becomes more complex and requires combining the various interpretations for a complete

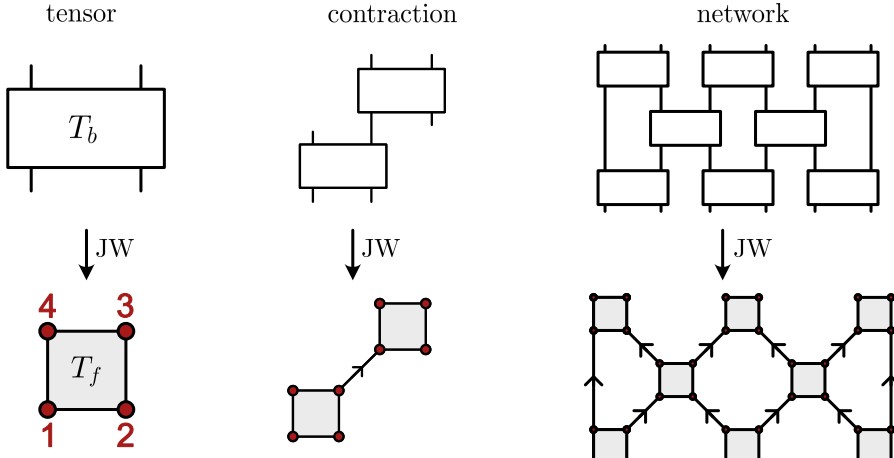

Figure 3: Mapping a planar TN of 2-qubit parity preseving gates to a fermionic Gaussian TN via JW transform, assigning an ordering of fermionic modes per tensor (right) and directions of index contractions (center). For a consistent choice of the latter two, the contracted bosonic TN evaluates to the same number as the fermionic TN (right).

understanding. The resulting phase diagram is shown in Fig. 2. For small values of $b$, the three phases remain separated by second-order phase transitions (solid lines). However, for small values of $a$ and comparably large values of $b$, the two topologically trivial (symmetry-breaking) phases are separated by a first-order transition (dashed line), with all three transition lines meeting in a tricritical point. As such, the phase diagram is topologically equivalent to that of a classical 2D *next-nearest-neighbor Ising* (NNNI) model [26, 27].

The NNNI model [28] and its cousin, the ANNNI [29] model, as well as various vertex models and the conditions under which they can be mapped to free fermionic models, have been studied extensively before [26, 27, 30–33]. Our model, however, offers the advantage of being a minimal model (with a particularly simple free fermion condition) that still displays the core phenomenology, while highlighting the intrinsic role of duality and topology. Furthermore, the formalization in terms of tensor networks allows for far-reaching solubility with a wide range of efficient numerical network methods [34].

The formalism in which our model is presented — a planar tensor network of fermion parity preserving tensors [35–38] — is rather general and, in particular, includes quantum circuits capable of universal quantum computation. Thus, the dualities discussed in the following establish connections between circuit-based quantum computation, statistical mechanics and interacting fermionic systems, and can be exploited to study the classical simulatability of quantum circuits [39]. It is the purpose of the current work to discuss these dualities in their most simple and quintessential fashion.

The rest of this work is structured as follows. In Sec. 2 we introduce the tensor network in its bosonic and fermionic version and discuss its interpretation as a fermionic system, and a statistical mechanics model – formulated as a vertex (loop gas) or, by duality, an Ising model. In Sec. 3 we derive the phase diagram by considering its statistical mechanics formulation as a loop gas and by comparing it to a classical Ising model with next-nearest neighbor interactions. Sec. 4 summarizes our findings and presents an outlook.

## 2 The model

We consider a two-dimensional tensor network on a square lattice. This geometric setup is equivalent to that of a brick wall quantum circuit (cf. Fig. 3). However, the tensors we consider are not unitary when interpreted as quantum gates. Nevertheless, we denote the map formed by the collection of all gates by $U$. For given initial and final conditions, $|\Psi_i\rangle, |\Psi_f\rangle$, the overlap $\langle\Psi_f|U|\Psi_i\rangle$ defines a fully contracted tensor network that evaluates to a number. Choosing the initial and final state to be the 'empty' product state vector $|\mathbf{0}\rangle := |0\rangle^{\otimes L}$, we define the bosonic partition sum $Z_b := \langle\mathbf{0}|U|\mathbf{0}\rangle$.

To enable a mapping to a fermionic tensor network [36, 37, 40–45], we restrict our discussion to parity-preserving two-qubit gates. In this setting, we can perform a Jordan-Wigner transformation [1, 16, 36, 40, 46, 47] of the entire TN, mapping the bosonic (spin) tensor network to a fermionic tensor network. Under this transformation, each bosonic tensor is mapped to a fermionic tensor. Specifically, we associate a fermionic mode, represented by a Grassmann variable $\theta^i$ with $\theta^0 = 1$, $\theta^1 = \theta$ to each index $i = 0, 1$ of the tensor, while choosing an ordering of fermionic modes. The index contractions (projections onto $\sum_i \langle i, i|$) are then translated to integrals over the two Grassmann variables associated with the corresponding bond, that is,

$$\int d\theta d\theta'(1 + \theta\theta') = \int d\theta d\theta' \exp\theta\theta'. \tag{4}$$

Note that this mapping also requires selecting an ordering for the modes involved in the contraction. We do this by assigning a direction to each contracted bond (cf. Fig.3).

Our goal is to assign the orderings in both steps such that the contraction of the fermionized tensor network matches that of the original tensor network, without introducing any residual sign factors. For a square lattice with 'empty boundary' ($|\Psi_i\rangle = |\Psi_f\rangle = |\mathbf{0}\rangle$), this can always be done by using the explicit assignments given in Ref. [16]. For different boundary conditions, the mapping needs to be augmented with a sign factor that depends solely on the boundary spins[1]. With this mapping in place, we arrive at a one-to-one correspondence of the respective tensor networks, allowing us to discuss the bosonic (spin) and fermionic versions in parallel.

The bosonic tensors are most naturally represented in terms two $2 \times 2$ matrices $u, v$, acting in the even and odd parity spaces, respectively. Meanwhile, the fermionic tensors are expressed as a sum of a Gaussian tensor and a residual quartic term. A Gaussian tensor is defined as an exponentiated Grassmann bilinear form determined by an anti-symmetric $4 \times 4$ matrix $A^\mathsf{T} = -A$, containing six parameters. The remaining two parameters are given by the normalization $N$ and the weight of the quartic term $b$. Together, this yields an identification between the parametrizations according to

$$T_b = \begin{pmatrix} u_{1,1} & & & u_{1,2} \\ & v_{1,1} & v_{1,2} & \\ & v_{2,1} & v_{2,2} & \\ u_{2,1} & & & u_{2,2} \end{pmatrix} \equiv N \begin{pmatrix} 1 & & & A_{1,2} \\ & A_{2,3} & A_{1,3} & \\ & A_{2,4} & A_{1,4} & \\ A_{3,4} & & & \mathrm{pf}A + b \end{pmatrix} \tag{5}$$

$$\Longleftrightarrow \quad T_f = N\, e^{\frac{1}{2}\theta^\mathsf{T} A\theta + b\theta_1\theta_2\theta_3\theta_4}.$$

By employing a global rescaling of the entire network, we normalize all tensors such that $u_{1,1} = N = 1$. This rescaling is inconsequential for our purposes, as it leaves quantities such as correlation functions invariant. In the remainder of this manuscript, we will focus on the choice $A_{i,j} = a\,\mathrm{sgn}(j - i)$, resulting in a two-parameter model defined solely by $a$ and $b$.

---

[1] We note that even with an unfavorable assignment of fermionic mode orderings, the resulting sign factors required to equate the contractions of the fermionic and bosonic tensor networks remain purely local. These factors are not detrimental to the efficient contraction of the tensor network in numerical algorithms [36]. For further details and alternative implementations of the JW transform on tensor networks, see Refs. [14, 15, 36–38, 47].

For any choice of boundary conditions, the fully contracted bosonic tensor network $Z_b$ evaluates to a number. To ensure the Jordan-Wigner transform takes its simplest form, we focus on 'empty' boundaries, i.e., the top and bottom layers of the brick-wall circuit are closed by projecting onto the state vector $|\mathbf{0}\rangle = |0\rangle^{\otimes L}$. In this case (combined with our choice of fermionic mode ordering), the contracted bosonic and fermionic tensor networks evaluate to the same number: $Z_b = Z_f$. If the tensor entries are positive and real, this number can be identified with a classical statistical mechanics partition function[2]. This allows us to discuss a *statistical mechanics interpretation* of the TN alongside its interpretation as a fermionic system (what we mean by 'system' here will be clarified momentarily).

## 2.1 Fermionic systems

Following the Grassmann variable formalism [16] for fermionic tensor networks, we write the contraction $\mathcal{C}$ of the fermionic tensor network as a Grassmann integral

$$Z_f = \int (\mathrm{d}\theta)_C \, e^{-S}, \quad S = -\frac{1}{2}\theta^\top(\oplus_x A + C)\theta - b\sum_x \theta_1^{(x)}\theta_2^{(x)}\theta_3^{(x)}\theta_4^{(x)}, \qquad (6)$$

where $C$ denotes the (directed) adjacency matrix of the square lattice (cf. Fig. 3) and $x$ labels lattice sites.

**Free fermions: Dirac Hamiltonian.** It is apparent that the tensor network corresponds to a free fermionic system if and only if $b = 0$. In this case, we can define the single particle (pseudo-) Hamiltonian $H := i(\oplus_x A + C)$ with the corresponding action $S_{\mathrm{free}} = \frac{i}{2}\theta^\top H\theta$. We can now interpret this Hamiltonian as a tight-binding model (albeit coupling to Majoranas instead of complex fermions), where the $A$-matrix defines hopping/pairing inside a unit-cell comprised of four Majoranas and $C$ corresponds to the unit-strength hopping/pairing between sites.

Referring to Ref. [16] for a detailed discussion of this Hamiltonian, we summarize here its most important features. Making use of translation invariance, we discuss the Hamiltonian in momentum space (see Appendix A for definitions). While any choice of $A$ yields a valid class D Hamiltonian, we focus on a particularly simple case, where $A_{i,j} = a \, \mathrm{sgn}(j - i)$. For generic real values of $a$, this four-band Hamiltonian is gapped. However, at $a = a_\pm := \sqrt{2} \pm 1$, we find a gap closing at $E = 0$ for $\mathbf{k}_- = (k_1, k_2) = (\pi, \pi)$ and $\mathbf{k}_+ = (0, 0)$, respectively. The band structure of the four-band Hamiltonian is shown in Fig. 4.

Close to the phase transition points, the Hamiltonian is well-described by the two-band approximations

$$H_\pm^{(2)}(q) = \sum_{a=1}^3 h_a(q)\sigma_a,$$

$$h_1 = -\frac{1}{2}\sin q_1, \quad h_{2,\pm} = -\frac{1}{2}(2 + m_\pm - \cos q_1 - \cos q_2), \quad h_3 = -\frac{1}{2}\sin q_2, \qquad (7)$$

where $q_i = k_i - k_{\pm,i}$ are the momenta relative to the gap closing points and $m_\pm = \pm 2(a - a_\pm)$ is the width of the energy gap at $q_1 = 0 = q_2$.

The Hamiltonian in Eq. (7) is known as the two-dimensional Haldane-Chern insulator [48]. Depending on the sign of $m$, it is found in one of two topologically distinct phases characterized by their Chern numbers. For $m > 0$ (i.e., $a_- < a < a_+$, the topological phase), the Chern

---

[2]If the entries are negative or complex, it may or may not be possible to transform the network to a network with only real and positive tensors by local gauge transformations. This is an interesting question in its own right but lies beyond the scope of this work.

number is $C = 1$ and for $m < 0$ (i.e., $a < a_-$ and $a > a_+$, the trivial phase), the Chern number is $C = 0$. Note that the two-band approximations above are valid only around $a_\pm$. In particular, at $a = 1$, there is an additional band crossing at $E = 1$ of the two lowest and two highest bands (cf. Fig. 4, middle). Hence, $H_+$ is not 'adiabatically connected' to $H_-$.

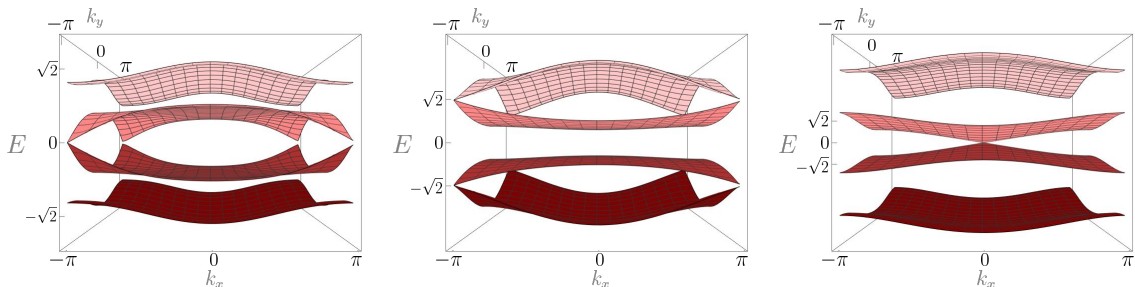

Figure 4: Band structure of the (four-band) pseudo-Hamiltonian $H = \mathrm{i}(\oplus A + C)$ for $A_{i,j} = a\,\mathrm{sgn}(j - i)$ at $a = a_-$ (left), $a = 1$ (center) and $a = a_+$ (right).

**Quartic terms.** The inclusion of quartic terms ($b \neq 0$) modifies the action in a manner reminiscent of interaction terms in condensed matter systems. However, the true path integral description of quantum many-body systems in terms of fermionic coherent states differs drastically from our setting, as its action features 'dynamic' quantum terms $\theta\,\partial_\tau\theta$, absent in our case.

While investigating the effects of strong coupling $b$ lies beyond the scope of this work, it is straightforward to argue that the system remains stable against weak non-linearities. To see this, we study the impact of the quartic terms in the vicinity of the phase transitions, where the two-band approximation is valid. Neglecting the effect of the higher bands altogether, we obtain a term that is strongly irrelevant on large length scales in the renormalization group sense. Starting from the representation of the quartic term in momentum space, we expand the fermionic modes $\theta_i(q)$ with momentum $q$ into the eigenbasis $v^\alpha(q)$ of $H(q)$ as $\theta_i(q) = \sum_\alpha v_i^\alpha(q)\eta_\alpha(q)$, where $\alpha = \pm 1, \pm 2$, and $\alpha = \pm 1$ denote the bands closest to zero energy. Restricting the summation to the latter two, we obtain

$$S_{\mathrm{int}} = b\sum_x \theta_1^{(x)}\theta_2^{(x)}\theta_3^{(x)}\theta_4^{(x)} \simeq b \sum_{q_1, q_2, q_3, q_4} \prod_{i=1}^{4} \sum_{\alpha_i = \pm 1} v_i^{\alpha_i}(q_i)\eta_{\alpha_i}(q_i)\,\delta_{q_1 + q_2 + q_3 + q_4, 0}\,. \quad (8)$$

Close to the phase transition, we now expand $v_i^\alpha(q) \simeq v_i^\alpha(0) + q\cdot\nabla v_i^\alpha(0)$. The zeroth order term vanishes as can be easily seen by its representation in real space

$$b\sum_x \prod_{i=1}^{4} \sum_{\alpha_i = \pm 1} v_i^{\alpha_i}\eta_{\alpha_i}^{(x)} = 0,$$

which contains at least two identical Grassmann numbers. Due to symmetry, the next non-vanishing order in gradients is the second order, containing two gradients. With the engineering dimension $[v] = L^{-1/2}$ determined by the quadratic gradient term, the quartic term dressed by two gradients scales as $\sim L^{-2}$, and thus becomes negligible at large length scales.

Going one step beyond the naive two-band approximation, we integrate out the higher bands as detailed in Appendix B. The result is a shift of the mass term in the Haldane-Chern Hamiltonian proportional to $b$. For small values of $b$, this mass shift alters the exact position of the phase transition, but does not lead to qualitative changes.

## 2.2 Statistical mechanics

**Vertex model.** We now turn to the interpretation of the contracted tensor network as the partition sum of a statistical mechanics model. Recall that all tensors are parity-even and identify the index value $i = 1$ with the presence of a string and the value $i = 0$ with its absence. This immediately results in a vertex or loop gas model, where the configurations in the partition sum are given by closed loops. In particular, the tensor in Eq. (5) specifies the weights of the $2^4/2 = 8$ string configurations around a vertex. For the choice $N = 1$ in Eq. (5), we obtain the weights depicted in Fig. 5.

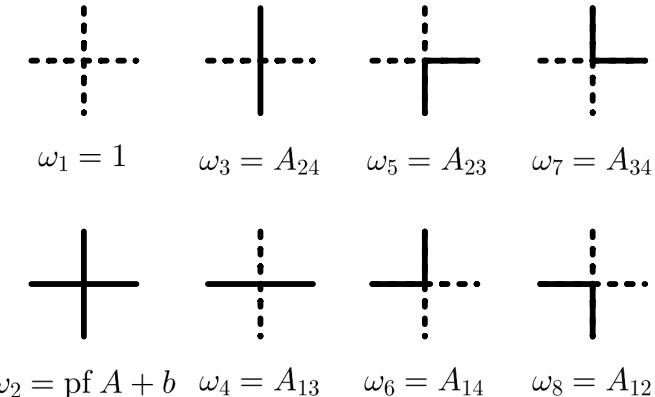

Figure 5: Weights of the vertex model (using established nomenclature [31, 32]) defined by the tensor network in Fig. 3 with tensors given in Eq. (5) and $N = 1$.

It is known [31, 32] that if the weights of a vertex model fulfill the *free fermion condition*

$$\omega_1\omega_2 + \omega_3\omega_4 = \omega_5\omega_6 + \omega_7\omega_8 \,, \tag{9}$$

the loop gas model can be mapped to free fermions. In our notation, this condition reduces to $b = 0$. We can now interpret the meaning of this condition in the language of statistical mechanics. If $b = 0$, the weight of the vertex configuration with four strings is given by a (signed) sum of all two-string configurations, reminiscent of a reading of Wick's theorem. In the particularly simple case where $A_{i,j} = a \, \text{sgn}(j-i)$, any two-string configuration is weighted by $a$, while a four-string configuration is weighted by $\text{pf}A = a^2$. Therefore, a self-intersection (crossing) has the same weight as two non-intersecting strings, and the total length is the only relevant factor determining the weight.

In contrast, if $b \neq 0$, crossings are either favored or disfavored depending on the sign of $b$. Defining the *crossing-weight* $c = 1 + b/a^2$, the partition function of the model is obtained by summing over all closed-loop configurations $l$ with weights $w(l)$, depending on the total string length $|l|$ and the number of intersections $|n_c|$ as

$$Z = \sum_{l:\text{closed strings}} w(l) \,, \quad \text{with} \quad w(l) = a^{|l|} c^{n_c} \,. \tag{10}$$

**Ising model.** Vertex models are known to be dual to (generalized) Ising models [49–51]. Following the seminal idea by Kramers and Wannier [52] (see also [53]) we can interpret the strings above as degrees of freedom in the *low-temperature expansion* of a (generalized) Ising model. Here, strings are interpreted as domain walls, which immediately leads us to an Ising model with the additional four-spin interaction $P_4$, favoring a checkerboard pattern around a plaquette (cf. Eq.(1)). Due to the presence of this purely quartic, four-spin interaction, we will from now on refer to this enriched Ising model as the *quartic Ising* (QI) model.

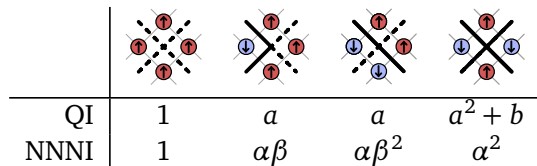

| | | | | |
|---|---|---|---|---|
| QI | 1 | $a$ | $a$ | $a^2 + b$ |
| NNNI | 1 | $\alpha\beta$ | $\alpha\beta^2$ | $\alpha^2$ |

Table 1: Weights of spin configurations of the QI and NNNI model. The weights of the configurations are invariant under $\pi/2$-rotations around the plaquette center and global spin flips.

We again focus on the homogeneous case $A_{i,j} = a \, \text{sgn}(j-i)$ and interpret the strings above as *domain walls* separating islands of opposite classical spins residing on the vertices of the dual lattice. We thus obtain a partition function where all four-spin configurations are assigned weights according to Tab. 1 (QI). For $b = 0$, we immediately obtain a nearest-neighbor Ising model where the coupling is (anti-) ferromagnetic for $a < 1$ ($a > 1$). For $b \neq 0$, the intersection of domain walls is energetically favored or penalized depending on the sign of $b$. To capture this effect, we need to enrich the nearest-neighbor Ising model with an additional four-spin interaction (cf. Fig. 1). We introduce the projector $P_4$ onto the two four-spin checkerboard patterns that correspond to a domain wall crossing (i.e., the last configuration in cf. Tab. 1 and its spin-inverted counterpart). With this, we obtain the partition function of the Ising model in terms of the two-spin and four-spin transfer matrices $T$ and $P$

$$T(a) = \begin{pmatrix} 1 & a \\ a & 1 \end{pmatrix}, \qquad P = c \, P_4 + (1 - P_4), \tag{11}$$

arranged in the tensor network in Fig. 6(a). Rescaling the partition function (which is equivalent to a global energy offset) and setting the inverse temperature $1/k_B T = 1$, we identify the Hamiltonian in Eq. (1) and visualized in Fig. 1.

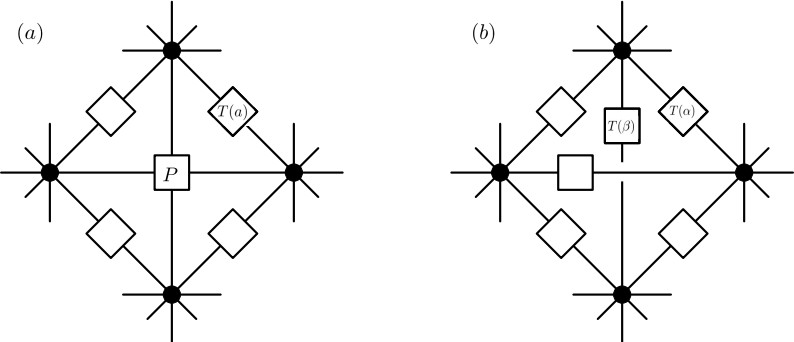

Figure 6: Partition sum of the QI model (left) and the NNNI model (right) in TN notation.

It is instructive to compare our model to the well-studied *next-nearest-neighbor Ising* (NNNI) model [26–28]. In this model, nearest and next-nearest neighbor interactions are determined by two parameters, $\alpha$ and $\beta$. We can then formulate its partition function as a tensor network composed of the transfer matrices $T(\alpha)$ and $T(\beta)$ (cf. Eq .(11)) according to Fig. 6(b). Again, the partition function is fully determined by the weights of the four-spin configurations, which are summarized in Tab. 1. We observe that there is no one-to-one correspondence between the two models. However, a comparison to the NNNI model allows us to infer the qualitative nature of the phase diagram, as discussed in detail at the end of the upcoming Sec. 3.2.

# 3 Phase diagram

In this section, we discuss the phase diagram of the model in the $(a, b)$-plane as presented in Fig. 2. For $a, b > 0$, there are three distinct phases. We first discuss them in the loop gas picture and then indicate their dual Ising phases.

## 3.1 Phase diagram in the loop gas picture

For small $a, b$ with fixpoint at $a = b = 0$, we find the phase called 'empty': the lattice devoid of strings. This corresponds to a ferromagnetic (F) Ising phase (no domain walls). For large $a, b$ with fixpoint at $a = b = \infty$, we find a loop-condensed phase called 'full', corresponding to a deeply *anti-ferromagnetic* (AF) checkerboard Ising phase (densely packed domain walls). For intermediate values of $a$ and small values of $b$ with fixpoint at $a = 1$, $b = 0$, we find the so-called 'topological' phase, where all closed-loop configurations are equally likely, reminiscent of the toric code ground state [17]. This corresponds to a *paramagnetic* (P) Ising phase [54]. In the fermionic picture, the 'empty' and 'full' configurations correspond to two distinct trivial insulators, while the 'topological' phase corresponds to a topological insulator.

The phase diagram along the $b = 0$ line is well-understood [16], and our main concern is the extension into the $b > 0$ regime (the case $b < 0$ is left for future work). As outlined in Sec. 2.1, the scaling analysis in the fermionic picture suggests that all three phases are robust against weak non-linearities given by small $b$. From the free model, we also know [16] that the two transitions at $a_\pm$ are of second order. Hence, the transition points at $b = 0$ smoothly extend to second-order transition lines for $b > 0$.

Concerning the qualitative behavior of these two lines, we find that, with increasing $b$, they drift towards smaller values of $a$. In the loop gas picture, this is consistent with the observation that a greater $b$ leads to a larger number of loop crossings, thus increasing the overall string content (cf. Eq. 10). In order to approximately preserve the weight profile of the configurations at the phase transition, the weight of the string segments $a$ needs to decrease.

The breakdown of the topological phase at some finite value of $b$ (for a fixed $a$) is expected. For any finite $a$, increasing the value of $b$ will eventually force the lattice to be fully covered by strings in order to maximize the number of loop crossings.

Focusing on $a \ll 1$, we find a first-order transition between 'empty' and 'full' phases at $b = 1$ for $a \to 0$. This third transition line implies the existence of a tricritical point where all three transition lines meet. This feature, as well as the nature of the $a = 0$, $b = 1$ transition, can be understood by comparing the Ising version of our model to the NNNI model discussed in Refs. [26–28].

## 3.2 Comparison with the Ising model with next-nearest-neighbor interactions

Notably, the Ising model with nearest- and *next-nearest-neighbor interactions* (NNNI) features the same three phases as our model. The partition functions of both models, expressed as tensor networks, are shown in Fig. 6, while Tab. 1 compares the weights of all possible four-spin configurations for our *quartic Ising* (QI) model and the NNNI model. Both models are invariant under cyclic permutations of the four spins around a plaquette as well as simultaneous spin flips of all four spins. Therefore, it suffices to consider the four configurations listed in the Tab. 1. Although the models appear similar, there is no one-to-one correspondence between them except at certain parameter points.

Let us examine the parameters of the two models and their relations in more detail. The parameter $\alpha$ is identical to our parameter $a$ and determines the nearest-neighbor interaction, where $\alpha < 1$ ($\alpha > 1$) corresponds to (anti-) ferromagnetic coupling. The parameter $\beta$ defines

the *next-nearest-neighbor* (NNN) strength, where again $\beta < 1$ ($\beta > 1$) means (anti-) ferromagnetic coupling. In the limit $\beta = 0$, $b = 0$, both models reduce to the *nearest-neighbor* (NN) Ising model and become identical. For finite $b$, $\beta$, there is no one-to-one correspondence between the two models; however, there are limiting cases where we expect the models to behave identically. Concretely, a strong anti-ferromagnetic NN coupling ($\alpha \gg 1$) combined with a ferromagnetic or sufficiently weak anti-ferromagnetic NNN coupling ($\beta \ll \alpha$) enforces a checkerboard pattern. The same occurs in the QI model in the limit $a \ll b$, when the quartic term dominates. This establishes the AF ('full') fixpoint. Similarly, if both interactions in the NNNI model are strongly ferromagnetic ($\alpha, \beta \ll 1$), there is an equivalence to the limit $a, b \ll 1$, establishing the F ('empty') fixpoint. For intermediate values, $\alpha, \beta \sim 1$, corresponding to $a \sim 1$, $b \ll 1$, we find the paramagnetic phase with its fixpoint at $b = 0$, $\beta = 0$ and $a = \alpha = 1$.

Having established the equivalence of the phases, we turn to the first-order transition. For the NNNI model, it is known [26, 55] that a first-order transition between F and AF occurs when, at sufficiently strong ferromagnetic NNN coupling, the NN coupling changes from F to AF. This can be seen clearly by considering the limiting case of infinitely strong ferromagnetic NNN coupling, $\beta = 0$. In this case, NNN bonds need to be aligned, enforcing two independent ferromagnetic phases on the two sublattices, given by the NNN couplings. The NN coupling then decides whether the two sublattices are either in the same state, leading to a uniform *ferromagnetic* (F) phase, or in different states, leading to a checkerboard pattern (AF). Comparing to the QI model, setting $\beta = 0$ corresponds to $b = \alpha^2$, $a = 0$. We know that the transition in the NNNI model happens at $\alpha = 1$ and, therefore, the transition in the QI model occurs at $b = 1$.

Regarding the tricritical point, we know that the NNNI model at $\alpha = 1$ decouples into two NNI models on the two sublattices defined by the NNN interaction. The individual models have a phase transition at $\beta = \beta_c = \sqrt{2} - 1$. This establishes the tricritical point. As discussed before, the parameters $\alpha = 1$, $\beta = \beta_c$ have no direct correspondence in the QI model, however, they are approximately matched by choosing $b = 1 - a_c^2 < 1$ and $0.17 \simeq \beta_c^2 < a_c < \beta_c^2 \simeq 0.41$. This places the tricritical point in the QI model towards the left of $a_-$, in agreement with the left-shift of the second-order line originating at $b = 0$, $a = a_-$.

In addition to the existence and approximate location of the transition lines and the tricritical point, we can infer the *renormalization group* (RG) flow of the parameters $a$ and $b$ in the vicinity of $b = 0$ from the scaling arguments in Sec. 2.1 and conclude that for $b \ll 1$, $b$ flows to zero. Together with the nature of the phase transition lines — separating the phases — and under the assumption that the tricritical point is completely unstable, this uniquely determines the topology of the RG flow lines as drawn in Fig. 2. Comparing to the RG flow lines in the NNNI model as worked out in Ref. [27], we observe topological equivalence between the two.

## 4   Conclusions and outlook

### 4.1   Summary

In this work, we have established a new kind of duality by considering a minimal extension of a matchgate tensor network beyond free fermions. While it has been known that a matchgate tensor network (corresponding to a free fermionic system) in two dimensions stands at the center of duality mappings between three systems of major significance in condensed matter physics — the toric code, the class D topological superconductor, and the two-dimensional Ising model [16] — it has been unclear how these dualities are affected by the inclusion of non-linearities. By introducing local non-linearities into the free tensor network, specified by

a single parameter, $a$, with the non-linearity strength modulated by another parameter, $b$, we have set up a rather simple two-parameter model. Moreover, the use of the tensor network as a bridge between fermionic matter, spin systems and vertex (loop gas) models allows for an efficient and versatile treatment of non-linearities in all three dual systems.

The rich physics of this minimal model is demonstrated by the phase diagram as discussed in Sec. 3. We find three distinct phases separated by first and second order phase transition lines, with all three coalescing at a tricritical point. The three phases allow interpretation from different perspectives: in the loop gas these are known as 'empty', 'topological' and 'full', in the Ising model as 'ferromagnetic', 'paramagnetic' and 'anti-ferromagnetic'. In the fermionic version of the system, the 'topological' or 'paramagnetic' phase is equivalent to the single-particle topological phase, while the other two exhibit a trivial band topology — as can be seen from inspecting their corresponding Chern numbers. In Sec. 2.1, we have applied scaling arguments to reason that the low-energy physics of the model remains robust to weak non-linearities, which is not surprising given the topological features of the system.

For stronger non-linearities, we use the statistical mechanics interpretation: considering the model in the loop gas picture as well as interpreting it as a two-dimensional Ising model with an additional quartic spin term, we draw comparisons with the *next-to-nearest neighbor Ising* (NNNI) model. In doing so, we establish the existence of a tricritical point in the phase diagram. The phase diagram itself is found to be topologically equivalent to that of the NNNI model.

## 4.2 Outlook

There are a number of natural extensions and generalizations of the model considered here, elevating the relevance of the discussions above beyond the particular system. For example, we can extend our model to include disorder and discuss a theory for ensembles of *random tensor networks* [56–58]. In this framework, a continuum field theory captures the ensemble-averaged theory. More concretely, we obtain the non-linear $\sigma$-model of the thermal quantum Hall effect, featuring a topological $\theta$-term. When allowing for different tensor network geometries, we further obtain a coupling of the fields to a background metric, allowing us to study the effect of a hyperbolic geometry [58].

Another interesting generalization involves restricting the tensors to unitary gates while allowing complex entries. In this setting, we can explore the connections between quantum circuits, statistical mechanics and fermionic systems. In particular, focusing on free fermionic tensors — or matchgate circuits — we can establish a correspondence between quantum circuits and non-Hermitian fermionic Hamiltonians. This correspondence, which likewise affords extension beyond the free fermion limit, is left for future research.

## Acknowledgments

We would like to thank Paul Fendley for illuminating discussions. This work has been supported by the Deutsche Forschungsgemeinschaft (DFG, German Research Foundation) under Germany's Excellence Strategy – Cluster of Excellence Matter and Light for Quantum Computing (ML4Q) EXC 2004/1 – 390534769 and within the CRC network TR 183 (Project Grant No. 277101999) as part of subproject B01, the BMBF (MuniQC-Atoms), and the European Research Council (DebuQC).

## A  Details on the Fourier transform

To derive the momentum space representation of the bilinear form $\theta^\mathsf{T} H \theta$ with $H = H_{\text{HC}}$ defined in Eq. (7), we use the following conventions and definitions. We take the Fourier transform with respect to the lattice structure $C$, meaning, we introduce

$$\theta_{q,i} = \frac{1}{L} \sum_\alpha e^{-iq\cdot\alpha} \theta_{\alpha,i}, \qquad \text{and} \qquad H_{i,j}(q,q') = \frac{1}{L^2} \sum_{\alpha,\alpha'} H_{\alpha,i;\alpha',j}\, e^{-iq\cdot\alpha + iq'\cdot\beta}, \qquad (12)$$

where $L^2$ is the number of cells in the lattice, $\alpha$ is a vector of unit-cell positions and $q = (q_1,q_2)^\mathsf{T}$ is its momentum conjugate. Since $H_{i,j}(q,q') = H_{i,j}(q)\delta_{q,q'}$ is diagonal in momentum space, we obtain the compact form $\sum_q \theta_{-q}^\mathsf{T} H(q)\theta_q$, where $H(q)$ is now a $4 \times 4$-matrix, known as the band Hamiltonian in condensed matter theory.

## B  Integrating out higher bands

Going one step beyond the two-band approximation, we consider the four-band model and integrate out the higher bands. That is, for all terms in Eq. (8) with two modes from high and two from low bands, we replace the higher modes by their expectation value as calculated from the free Hamiltonian $H^{(2)}$

$$\eta_q^{s2}\eta_{q'}^{s'2} \mapsto \left\langle \eta_q^{s2}\eta_{q'}^{s'2} \right\rangle = \frac{\varepsilon_{s,s'}}{i\epsilon_2(q)}\delta_{q,-q'}, \qquad (13)$$

where $s,s' = \pm$ and $\epsilon$ denotes the anti-symmetric tensor. With this, we obtain

$$S_{\text{int}} \simeq \frac{ib}{4}\varepsilon_{i,j,k,l} I^{i,j} \sum_{q'} v_{s,1}^k(q') v_{s',1}^l(-q') \eta_{q'}^{s,1}\eta_{-q'}^{s'1}, \qquad (14)$$

where

$$I^{i,j} = \varepsilon_{s,s'}\int_{\text{BZ}} \frac{d^2q}{(2\pi)^2}\epsilon_2(q)^{-1} v_{2,s}^i(q) v_{2,s'}^j(-q). \qquad (15)$$

The resulting action is quadratic in the low-lying modes $\eta^1$ and, writing $\eta = (\eta^{1,+}, \eta^{1,-})^\mathsf{T}$, it can be brought to the form

$$S_{\text{int}} = \frac{i}{2}\sum_q \eta_{-q}^\mathsf{T}\Big(\epsilon_1(q)\sigma_2 + \delta m(q)\sigma_2\Big)\eta_q, \quad \text{with} \quad \delta m(q) = \frac{b}{2}\epsilon_{i,j,k,l} I^{i,j} v_+^k(-q) v_-^l(q). \qquad (16)$$

For $q$ small, $\delta m(q) \simeq \delta m(0) + \mathcal{O}(q^2)$, while $\epsilon_1(q) \simeq |q|$. Therefore, up to linear order in $q$, the low-lying bands are only shifted by a constant $\delta m(0)$. In conclusion, the effect of the quartic term is to shift the mass of the Haldane-Chern Hamiltonian by a factor $\delta m(0)$.

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
