# Peer review of "A minimal tensor network beyond free fermions"

_SciPost Physics_

## Round 1 · Referee Report · Anonymous (Referee 1) · 2025-2-23

Strengths

Mostly well written
I think the work is scientifically valid and a novel technique

Weaknesses

I'm not clear on how broad the interest will be. This may be my ignorance though.

Report

The manuscript by Wille, Usoltcev, Eisert, and Altland is a
demonstration that the dualities that exist between Toric Code, Ising, and (noninteracting) fermionic superconductors can be generalized to the interacting non-linear case. The authors write a model that in the Ising language involves four-spin interactions, and converts it into the fermionic language which then includes four fermion terms. The analysis of the problem in this language seems a bit more natural and allows the authors to show the power of the transformation.

Overall I think the work is a valid calculation. The results do not seem particularly surprising, but the main point appears to be to demonstrate the technique rather than to find any surprising results. I suspect the same general results could be obtained with other perturbative approaches just as well.

My main concern is how large the audience will be for this work. I
agree it rises to the level of SciPost core, although perhaps not
SciPost Physics. Since I do not work closely in this field, maybe I
don't have the best judgement of how many people wil be interested. If other referees think it rises to SciPost Physics, I won't object.

There are a few things that might be detailed more in this work:
possibly in an appendix so as not to interrupt the flow of an
otherwise fairly streamlined paper. For example, some information about the mapping from fermionic to bosonic would be very welcome. Even looking up Ref. 16 by the same authors is not very helpful. Since this is a key step it would be nice to see it in detail.

Another piece that could be explained better is the details of the
phase diagram, fig 2. Certain pieces of this diagram have been
actually calculated, but other pieces seem to be just guess. For
example, at the tricritical point, the full/topological boundary is
guessed to have positive slope, but I don't think there is actually
evidence of this?

Overall I would recommend the authors consider the optional additions mentioned here.

Requested changes

Consider optional additions.

Recommendation

Publish (meets expectations and criteria for this Journal)

---

## Round 1 · Referee Report · Anonymous (Referee 2) · 2025-3-6

Strengths

1- The paper introduces a new duality between classical spin systems and interacting fermions, in connection to circuit-based quantum computation and tensor network formulation
2- The paper is well written and clear description of the connection to the 2D nearest-neighbor Ising model
3- It offers a promising basis for future research

Weaknesses

1- A quantitative, numerical study of the phase diagram using tensor network methods is not provided and would have been an interesting addition

Report

In this paper, the authors establish a duality between classical spin systems and fermionic systems for a simple model of interacting fermions in two dimensions based on a tensor network formalism, extending the case of free fermions. The model is described by two parameters that control the intra-unit-cell hopping strength (corresponding to nearest-neighbor spin interactions in the classical case) and the strength of a four-fermion term (four-spin interaction). The authors study the phase diagram of the model and make connection to the classical 2D next-nearest-neighbor Ising model, which exhibits a topologically equivalent phase diagram, featuring a single particle topological phase (paramagnetic phase), two trivial band insulators (ferromagnetic and antiferromagnetic phase), and a tricritical point. They also discuss the phases in terms of the loop gas picture of a corresponding vertex model.

Dualities in statistical physics play an important role in statistical physics, and in my opinion, this paper provides an interesting and significant contribution in this field, offering new connections between statistical mechanics and interacting fermionic systems, as well as circuit-based quantum computation, expressed in a tensor network language. The paper is well written, and it also offers an interesting basis for future research, as the authors outline in their outlook.

For these reasons, I recommend publication of this paper in Scipost Physics.

I have only minor suggestions for improvements:
1) It would help to mention in the caption of Fig. 1 that b controls the strength of the four fermion ring exchange term.
2) Define the acronym of the ANNNI model
3) At the bottom of page 11, rephrase the sentence: "By introducing local non-linearities into the free tensor network, specified by a single parameter, a, with the non-linearity strength modulated by another parameter, b, we have set up a rather simple two-parameter model.", since one may misread it as that the local non-linearities are "specified by a single parameter, a".

Recommendation

Publish (meets expectations and criteria for this Journal)

---

## Round 1 · Referee Report · Anonymous (Referee 3) · 2025-3-21

Strengths

1 - A duality from a spin partition function to free fermions is extended to include interactions in a controlled way.
2 - Tensor network approach is flexible and allows connections to be made to partition functions of spin/loop gas systems, interacting fermions, and parity-preserving quantum circuits.

Weaknesses

1 - The work is restricted to a specific model, although is based on a very general idea based on duality. It would be interesting to see how far these ideas can generalize to other 2D spin models.

Report

This work generalizes a well-known duality between the partition function of a 2D spin system and a system of free fermions to include interactions. This uncovers a rich phase diagram with gapped phases separated by critical lines that meet at a tricrtitical point.

A tensor network approach is taken to formulate the models in question, and their duality. This allows connections to be made between partition functions of an Ising model or dual loop gas, interacting fermions, and parity-preserving quantum circuits.

The work is clearly explained and makes some appealing connections. I find it likely that this work will inspire follow up studies in this direction. For these reasons I recommend it for publication.

Requested changes

The work can be published as is.

Recommendation

Publish (meets expectations and criteria for this Journal)

---

## Editorial Decision

awaiting_resubmission